# Allogeneic Stem Cell Transplantation in Multiple Myeloma: Risk Factors and Outcomes in the Era of New Therapeutic Options—A Single-Center Experience

**DOI:** 10.3390/cancers15245738

**Published:** 2023-12-07

**Authors:** Irene Strassl, Alexander Nikoloudis, Sigrid Machherndl-Spandl, Veronika Buxhofer-Ausch, Michaela Binder, Dagmar Wipplinger, Olga Stiefel, Emine Kaynak, Robert Milanov, Christoph Aichinger, Stefanie Nocker, Thomas Bauer, Stefanie Kreissl, Michael Girschikofsky, Andreas Petzer, Ansgar Weltermann, Johannes Clausen

**Affiliations:** 1Division of Hematology with Stem Cell Transplantation, Hemostaseology and Medical Oncology, Department of Internal Medicine I, Ordensklinikum Linz, Fadingerstrasse 1, 4020 Linz, Austria; alexander.nikoloudis@ordensklinikum.at (A.N.); sigrid.machherndl-spandl@ordensklinikum.at (S.M.-S.); veronika.buxhofer-ausch@ordensklinikum.at (V.B.-A.); dagmar.wipplinger@ordensklinikum.at (D.W.); olga.stiefel@ordensklinikum.at (O.S.); emine.kaynak@ordensklinikum.at (E.K.); robert.milanov@ordensklinikum.at (R.M.); christoph.aichinger@ordensklinikum.at (C.A.); stefanie.nocker@ordensklinikum.at (S.N.); thomas.bauer@ordensklinikum.at (T.B.); stefanie.kreissl@ordensklinikum.at (S.K.); michael.girschikofsky@ordensklinikum.at (M.G.); andreas.petzer@ordensklinikum.at (A.P.); ansgar.weltermann@ordensklinikum.at (A.W.); johannes.clausen@ordensklinikum.at (J.C.); 2Medical Faculty, Johannes Kepler University Linz, Altenberger Strasse 69, 4040 Linz, Austria

**Keywords:** multiple myeloma, allogeneic stem cell transplantation, graft-versus-myeloma effect, plasma cell leukemia, haploidentical, bispecific antibody

## Abstract

**Simple Summary:**

Patients with multiple myeloma (MM) refractory to conventional treatment strategies represent an unmet medical need. Allogeneic stem cell transplantation (allo-HSCT) is a controversially discussed treatment option in MM, only used in selected patients due to its high rates of morbidities and mortality. We present a retrospective analysis of all MM patients who underwent allo-HSCT at our center during the last 10 years. In the overall cohort, and especially in patients with at least VGPR prior to allo-HSCT, remarkable long-term survival is possible. Therefore, even in the context of new treatment modalities, allo-HSCT may still offer a therapeutic option for individual MM patients.

**Abstract:**

Background: Despite major treatment advances, multiple myeloma remains incurable. The outcome of patients who are refractory to immunomodulatory agents, proteasome inhibitors, and anti-CD38 monoclonal antibodies is poor, and improved treatment strategies for this difficult-to-treat patient population are an unmet medical need. Methods: This retrospective, unicentric analysis included 38 patients with relapsed/refractory multiple myeloma or plasma cell leukemia who underwent allogeneic stem cell transplantation (allo-HSCT) between 2013 and 2022. Survival outcomes, relapse incidence, and non-relapse mortality were calculated according to remission status, date of allo-HSCT, cytogenetic risk status, timing, and number of previous autologous HSCTs. Results: The median PFS was 13.6 months (95% CI, 7.7–30.4) and the median OS was 51.4 months (95% CI, 23.5–NA) in the overall cohort. The cumulative incidence of relapse at 3 years was 57%, and non-relapse mortality was 16%. The median PFS and OS were significantly longer in patients with very good partial remission (VGPR) or better compared to patients with less than VGPR at the time of allo-HSCT (mPFS 29.7 months (95% CI, 13.7–NA) vs. 6.5 months (95% CI, 2.6–17.0); *p* = 0.009 and mOS not reached vs. 18.6 months (95% CI, 7.0–NA); *p* = 0.006). Conclusion: For selected patients, allo-HSCT may result in favorable overall survival, in part by providing an appropriate hemato-immunological basis for subsequent therapies.

## 1. Introduction

Despite tremendous advances in the treatment of multiple myeloma (MM) over the past two decades, the disease remains mostly incurable [1]. The introduction of immunomodulatory drugs (IMiDs), proteasome inhibitors (PIs), and anti-CD38 monoclonal antibodies (anti-CD38 Abs) has led to a significant improvement in the outcome of patients with MM [2]. Of particular concern is the subset of patients who are refractory to IMiDs, PIs, and anti-CD38 Abs, showing early relapse after first- or second-line therapy, including autologous stem cell transplantation (ASCT) or primary refractory MM patients. These difficult-to-treat patient populations have particularly dismal outcomes, frequently associated with poor overall survival [3,4]. Allogeneic hematopoietic stem cell transplantation (allo-HSCT) represents a treatment option for MM patients with early relapse to other established MM therapies, especially after ASCT, or lacking further therapeutic options after a long-standing disease history [5]. Nevertheless, the role of allo-HSCT in the treatment of MM remains challenging due to its high toxicity along with limited response rates, although it is a potentially curative approach for a limited number of selected patients [6]. Regardless of the ongoing controversial debate about the indication for allo-HSCT in MM, the use of allo-HSCT is increasing in Europe even after the introduction of new therapeutic agents, reflecting the high unmet medical need for further treatment options in relapsed/refractory MM (RRMM) [7]. The adaptation of conditioning regimens with a shift from myeloablative conditioning (MAC) to reduced-intensity conditioning (RIC) resulted in a significantly reduced non-relapse mortality (NRM) but higher relapse rate [8]. In a proportion of patients, allo-HSCT following RIC can still lead to a long-term response through a graft-versus-myeloma (GvM) effect. The long-term follow-up of a large, pooled analysis of patients treated with tandem autologous versus autologous–allogeneic transplantation showed significantly longer overall survival with ASCT/RIC-allo-HSCT compared to tandem ASCT [9]. Importantly, median post-relapse survival was also significantly improved by allo-HSCT, underlining a durable GvM effect leading to improved efficacy and disease control even with subsequent therapies. A repeated and similarly good response to therapy regimens already used before allo-HSCT was also observed in other studies as well as deep and durable responses to the current standard-of-care therapeutics like IMiDs, PIs, and anti-CD38 Abs with good tolerability [9,10,11,12]. Data on the efficacy and safety of BCMA-targeted therapies including T-cell-recruiting agents after allo-HSCT are sparse and limited to a few patients [13,14], in part due to the frequent exclusion of patients post allo-HSCT from randomized trials.

We conducted a retrospective analysis including all patients who underwent allo-HSCT in the last 10 years at our center to evaluate the outcomes of allo-HCST, identify factors that determine the outcome, and document the subsequent course after MM relapse.

## 2. Materials and Methods

### 2.1. Patient Population and Data Collection

This single-center retrospective analysis included all consecutive patients diagnosed with MM or plasma cell leukemia who underwent allo-HSCT between November 2013 and November 2022. Data cut-off for follow-up data was 14 August 2023. All patients provided written informed consent to data collection and analysis in compliance with the European Society for Blood and Marrow Transplantation (EBMT), national authorities, and the institutional review board/local ethics committee. Data collected included patient demographics, disease characteristics, treatment history, response status, transplant details, outcome data, and data on transplant complications and NRM. International Staging System (ISS) [15] was documented at time of initial diagnosis as well as cytogenetic risk profile. Besides high-risk status, defined by the International Myeloma Working Group (IMWG) as including t(4;14), t(14;16), and del(17p) [16], the gain or amplification of chromosome 1q and complex karyotypes were recorded. Data of all therapy regimens before and after allo-HSCT as well as number and time course of previous ASCTs are available. Response assessment was performed using the IMWG consensus criteria for response and the minimal residual disease assessment in MM 2016 [17]. MM response was assessed after the last therapy regimen prior to allo-HSCT. In addition, the best responses to allo-HSCT and to post-allo-HSCT relapse therapy were collected. Conditioning regimens were classified as “myeloablative” and “reduced-intensity” according to published criteria [18].

### 2.2. Statistical Analysis

Descriptive analyses were performed to summarize medians with the minimum and maximum (range) for continuous variables and counts and percentages for categorical variables. Progression-free survival (PFS) and overall survival (OS) were estimated using the Kaplan–Meier method. PFS and OS were measured from the date of allo-HCST until progression or death, respectively. In addition, OS was evaluated from the first relapse after allo-HSCT. Cumulative incidence functions of relapse and NRM were calculated. Non-relapse mortality was considered a competing risk for relapse; relapse was considered a competing risk factor for NRM. Gray’s test was performed to compare cumulative incidence between groups. To estimate cause-specific hazard ratios (HRs), Cox regression models were used. Two-sided *p* values < 0.05 were considered statistically significant.

All statistical analyses and visualizations were carried out using the R statistical software version 4.2.2 [19].

## 3. Results

### 3.1. Patient, Disease, and Transplant Characteristics

Between 13 November 2013 and 2 November 2022, 38 consecutive patients with RRMM received an allo-HSCT at our center. The patients’ baseline characteristics are shown in Table 1. The median age at the time of allo-HSCT was 55 years (range, 38–67), and 61% of patients were male. The median time from the initial diagnosis of MM to allo-HSCT was 3.8 years (range, 0.3–13.6). The proportion of patients with any high-risk characteristic was remarkable, with 53% having one or more high-risk cytogenetic abnormality (CA) including t(4;14), t(14;16), del(17p), gain or amp 1q, or a complex karyotype; and 21% of patients had extramedullary disease (EMD) prior to allo-HSCT. Comparing the two cohorts of patients with allo-HSCTs from 2013 to 2018 and from 2019 to 2022, those transplanted from 2019 to 2022 were slightly older and more likely to have high-risk features such as a high-risk CA or EMD. All but two patients had undergone one or more ASCTs (95%); in both patients without ASCT, autologous cell collection was not possible due to insufficient treatment response. Half of the patients had two ASCTs, and 13% of the ASCTs were preplanned tandem transplantations due to high-risk MM. Patients were heavily pretreated with a median of seven previous therapy regimens (range, 4–13). A total of 74% of patients were triple-class-exposed, i.e., had received at least one IMiD, one PI, and one anti-CD38 A; and 24% were triple-class refractory. Since the initial approval of an anti-CD38 Ab did not occur until mid-2016, the proportion of triple-class-exposed patients in the cohort transplanted between 2013 and 2018 was rather low at 47%. In contrast, all patients transplanted as of 2019 were triple-class-exposed prior to allo-HSCT.

Details of the allo-HSCTs including the stem cell and donor characteristics, conditioning regimen, and GVHD-prophylaxis are depicted in Table 2. The source of stem cells was peripheral blood in 84% of patients; all patients transplanted after 2018 received a graft generated from peripheral-blood stem cells. Conditioning regimens were myeloablative in 55% of patients and reduced-intensity in 45%. The intensity of conditioning was adjusted according to previous therapies and patient fitness. Total body irradiation (TBI) between 2 and a maximum of 8 Gray was part of the conditioning regimen in 58% of patients; the proportion of TBI-based conditioning was more frequent in transplantations after 2018 at 74% than in earlier years at 42%. Donors were divided into matched related donors (MRD), matched and mismatched unrelated donors (MUD and MMUD), and haploidentical related donors (Haplo). The distribution was 26% MRD, 34% MUD, 8% MMUD, and 32% Haplo. GVHD prophylaxis differed according to donor type, and post-transplant cyclophosphamide/tacrolimus/mycophenolate mofetil (PTCy-Tac-MMF) was used after Haplo transplantation and in a proportion of MMUDs. The majority of other patients received a combination of a calcineurin inhibitor and MMF. Anti-T lymphocyte globulin (ATLG) was used in 66% of transplants, particularly in HLA-matched HSCT.

The overall response rate (ORR) to the last therapeutic regimen before allo-HSCT, defined as achieving at least partial remission (PR), was 87%; 47% of patients achieved very good partial remission (VGPR) or better, 18% of patients achieved complete remission (CR), and 5% achieved stringent complete remission (SCR).

### 3.2. Response to Allo-HSCT and Maintenance

After allo-HSCT, the depth of remission improved in a large proportion of patients, with 55% of all patients achieving SCR. In contrast, the proportion of patients with an inadequate response prior to allo-HSCT remained largely unchanged after allo-HSCT, with 13% versus 11% of patients with less than PR (Figure 1).

Patients transplanted between 2019 and 2022 were more likely to receive maintenance therapies (47%) than patients with allo-HSCTs before 2019 (5%). Mostly PIs or IMiDs or combinations were used as maintenance therapies. Overall, the rate of maintenance therapy after allo-HSCT was relatively low at 26%. The primary reason to withhold maintenance therapy was preexisting or expected toxicity; in the case of PIs, mainly preexisting polyneuropathy; in the case of IMiDs, the risk of inducing GVHD; and in the case of both drug classes, severe and prolonged cytopenia. On the other hand, 37% of patients received donor lymphocyte infusions (DLI), either in preemptive (minimal residual disease, lack of SCR, or declining chimerism) or therapeutic indication (after relapse) in the further course.

### 3.3. Survival Outcomes

After a median follow-up of survivors of 37.5 months (95% CI, 21.8–57.4), the median PFS following allo-HSCT was 13.6 months (95% CI, 7.7–30.4) in the overall cohort. The 3-year PFS probability was 27.1% (95% CI, 0.15–0.48). The median OS was 51.4 months (95% CI, 23.5–NA), with an OS probability of 53.1% (95% CI, 0.38–0.74) at 3 years (Figure 2).

The median PFS was significantly longer in patients with VGPR or better compared to that of patients with less than VGPR at the time of allo-HSCT (29.7 months (95% CI, 13.7–NA) vs. 6.5 months (95% CI, 2.6–17.0); HR 0.38; 95% CI, 0.18–0.81; *p* = 0.009) (Figure 3A). The median OS was not reached in patients with VGPR or better and was significantly longer than that in patients with less than VGPR (NR vs. 18.6 months (95% CI, 7.0–NA); HR 0.23; 95% CI, 0.08–0.72; *p* = 0.006) (Figure 3B).

In the cohort of patients transplanted between 2019 and 2022, compared to patients transplanted between 2013 and 2018, no significant difference was revealed for PFS (median 17.0 months (95% CI, 10.7–NA) vs. 13.6 months (95% CI, 4.8–48.3); HR 0.80; 95% CI, 0.37–1.74; *p* = 0.58), while OS (median NR (95% CI, 28.9–NA) vs. 24.1 months (95% CI, 13.6–NA); HR 0.45; 95% CI, 0.16–1.30; *p* = 0.13) tended to be longer. The presence of at least one high-risk CA including t(4;14), t(14;16), del(17p), gain/amp 1q, or complex karyotypes resulted in significantly worse OS compared with patients without high-risk CA (median 24.1 months (95% CI, 16.3–NA) vs. NR (95% CI, 51.4–NA); HR 0.28; 95% CI, 0.08–1.04; *p* = 0.04). The PFS results according to high-risk status were borderline significant (*p* = 0.05) (Figure 4). The time interval between the last ASCT and allo-HSCT affected OS but not PFS. The 3-year OS probability was significantly lower in patients who underwent allogeneic transplantation less than 2 years after the last ASCT than in patients who underwent ASCT more than 2 years before their allo-HSCT (38% (95% CI, 0.20–0.71) vs. 67% (95% CI, 0.47–0.97); *p* = 0.02). Finally, patients with one or two ASCTs before their allo-HSCT were compared. No statistically significant difference in PFS and OS was seen, but there was a trend towards worse outcome in patients with two ASCTs before allo-HCST. The number of treatment regimens administered prior to allo-HSCT did not have a significant impact on OS (≤6 versus >6 regimens; 3-year OS 64% versus 48%; *p* = 0.70). Neither the type of conditioning (*p* = 0.60) nor the type of donor (*p* = 0.44) had a significant influence on OS.

Multivariate analysis confirmed the significant impact of remission status (less than VGPR versus at least VGPR before allo-HSCT) and cytogenetic high-risk factors (any high-risk CA versus no high-risk CA) on overall survival (Table 3).

### 3.4. Relapse Incidence, GHVD and Non-Relapse Mortality

The cumulative incidence of relapse at 3 years was 57% (95% CI, 0.38–0.72), and the cumulative incidence of NRM at 3 years was 16% (95% CI, 0.06–0.30) in the overall cohort (Appendix A). The most common causes of NRM were infections and acute GVHD (aGVHD). The cumulative incidence of aGVHD until day 100 after allo-HSCT was 42.1% (95% CI, 0.26–0.57) for aGVHD grade 2-4 and 18.4% (95% CI, 0.08–0.32) for aGVHD grade 3-4. For moderate/severe chronic GVHD (cGVHD), the 2-year cumulative incidence was 12.5% (95% CI, 0.03–0.26).

Depth of remission before allo-HSCT had no significant impact on the cumulative incidence of relapse, with a 3-year incidence of 53% (95% CI, 0.23–0.77) in patients with VGPR or better, compared to 60% (95% CI, 0.34–0.79) in patients with allo-HSCT with less than VGPR (*p* = 0.21). By contrast, cytogenetic high-risk status tended to have an adverse impact on the cumulative relapse incidence, with a 3-year incidence of 65% (95% CI, 0.36–0.84), compared to 40% (95% CI, 0.13–0.66) in patients without a high-risk CA (*p* = 0.08) (Figure 5). The date of allo-HSCT, the interval between the last ASCT and allo-HSCT, and the number of ASCTs did not affect the cumulative incidence of relapse. 

NRM was non-significantly lower in patients with at least VGPR at the time of allo-HSCT in patients transplanted between 2019 and 2022 and in patients with a longer interval between their last ASCT and their allo-HSCT (≥2 years) (Figure 6).

### 3.5. Subsequent Therapies and Outcomes after Relapse Post-Allo-HSCT

Overall, 58% of all patients relapsed after allo-HSCT by the time of the data cut-off. A total of 45% of patients have died, including 63% of the patients transplanted before 2019 and 26% of the patients transplanted from 2019 to 2022. The most common cause of death was the progression of MM, at 53%. All but one of the twenty-two patients with relapse post allo-HSCT received at least one subsequent treatment line. Appendix A shows the best response achieved to any subsequent therapy regimen after relapse post-allo-HSCT.

The estimated median OS from the first relapse after allo-HSCT was 22.6 months (95% CI, 13.7–NA) (Appendix A). Of the surviving 21 patients, 16 are in ongoing remission (76%). The median number of therapy regimens (including both maintenance/consolidation and salvage therapies) post-allo-HSCT in the overall cohort was two (range, 0–9). Relapsed patients received a median of four therapy regimens after allo-HSCT (range, 0–9). Eleven (50%) of the twenty-two relapsed patients were able to achieve at least a VGPR on a subsequent line of therapy, and six patients (27%) achieved a SCR. A total of 41 different therapy regimens were used to treat RRMM post-allo-HSCT. A summary of the different substances and substance classes that were administered is shown in Appendix A.

In terms of treatment regimens administered for relapse after allo-HSCT, IMiDs were used most frequently in a total of 18 patients (82%). The most commonly administered IMiD was pomalidomide in 15 patients (68%). A total of 17 patients (77%) received carfilzomib, 12 of whom also received another PI during the further course of treatment. Treatment with monoclonal antibodies was also common in relapsed patients after allo-HSCT at 77%. Five patients with t(11;14) received venetoclax combinations at relapse post-allo-HSCT. BCMA-targeted therapies, specifically, belantamab mafodotin and teclistamab, were used upon availability in the case of relapse. Five patients (23%) received belantamab mafodotin and a total of seven patients (32%) were treated with bispecific antibodies; all received teclistamab, and one patient additionally received talquetamab. None of these patients developed unexpected side effects, and importantly, no patient developed GVHD. Six of the seven patients treated with teclistamab had extramedullary MM manifestations at the time of therapy initiation, and one patient with plasma cell leukemia had an incipient biochemical relapse. Three patients showed ongoing responses at the data cut-off for 15.6, 12.1, and 8.6 months. 

## 4. Discussion

For many years, allo-HSCT has been a controversially discussed topic in multiple myeloma. With the introduction of today’s standard of care in earlier lines of therapy using different combinations of PIs, IMiDs, and anti-CD38 Abs, the prognosis for a large proportion of MM patients has improved significantly, both in newly diagnosed MM patients and also in the relapsed/refractory setting. However, there remains a small proportion of mostly young patients who relapse despite treatment with all available conventional therapy options. This patient population represents an unmet medical need, and to date, allo-HSCT has been a potential treatment modality that has been able to provide prolonged survival for a subset of these difficult-to-treat patients. The present analysis of 38 consecutive patients who underwent allo-HSCT at our center during the last 10 years shows a remarkable median OS of 51.4 months in heavily pretreated patients after a median of seven prior therapy regimens. In addition, for allo-HSCTs performed between 2019 and 2022, the median OS was not reached. The median PFS of 13.6 months in the context of the evaluated population is also notable, particularly the median PFS of 17 months for the allo-HSCTs performed between 2019 and 2022. Our results appear to be favorable compared with historical data from a large EBMT analysis. Specifically, for patients transplanted after 2004 in later lines of therapy, the EMBT analysis showed a median OS of 26 months and a median PFS of 11 months [7]. Even better outcomes have been observed in our analysis if at least VGPR was achieved prior to allo-HSCT, with a median PFS of 29.7 months, a median OS not reached, and a 3-year survival probability of 72%. These results are clearly superior to various real-world evaluations of triple-exposed MM patient populations treated with standard-of-care therapies, where durations of response or time to next treatment were well below one year and overall survival was between 8 and 15 months [3,20,21,22,23,24,25]. Considering the deepest possible remission before allo-HSCT to be the most important factor for a good outcome, the indication for allo-HSCT should be restricted to patients who are able to achieve at least VGPR to bridging-therapy before allo-HSCT. Patients who achieve less than a PR prior to planned allo-HSCT do not appear to benefit from this treatment modality and should be managed with an alternative therapy. Other factors such as cytogenetic and clinical high-risk features like early relapse after ASCT and EMD should be considered as well, as they may lead to a deterioration of outcome. The presence of at least one high-risk genetic abnormality consisting of t(4;14), t(14;16), del(17p), gain/amp 1q, or complex karyotypes resulted in significantly worse OS in this analysis, whereas the 3-year OS in patients without high-risk features was excellent at 85.7%. Thus, the adverse impact of high-risk cytogenetic abnormalities could not be overcome by allo-HSCT in this analysis.

Independent of disease-associated factors, achieving the lowest possible rate of transplant-related complications and NRM is also of importance. Prior MM therapies and those potentially needed in the future should be included in considerations regarding the selection of conditioning regimen and GVHD prophylaxis. The rather low rate of cGVHD in our analysis may be explained by maintenance or relapse therapies in a relevant proportion of patients. In addition to steroid-containing therapies, PIs and anti-CD38 Abs are supposed to have preventive effects on GVHD. Overall, the cumulative incidence of NRM in this analysis was relatively low at 16% at 3 years. Among more recently transplanted patients, the estimated 3-year NRM was even lower at 11%.

Limitations of this analysis are the retrospective design and the small number of cases. These factors reflect the fact that larger prospective analyses of allo-HSCT in RRMM are scarce and that this treatment modality is reserved for a small proportion of patients for whom other options are not available.

Although relapses after allo-HSCT are frequent, a response can usually be achieved by a thoughtful use of previously administered therapies. The observed long median OS (NR for allo-HSCTs between 2019 and 2022), as compared to the shorter median PFS (17 months in the above cohort) in our analysis, suggests that therapies that were ineffective before allo-HSCT are able to induce remissions again in the context of a reestablished (donor-derived) immune system. In addition, novel therapeutic options are available, most notably, chimeric antigen receptor T (CAR-T) cell therapies and bispecific antibodies. To date, few experience has been gained with CAR-T cells after allo-HSCT, though existing data show comparable efficacy and good tolerability in patients after allo-HSCT [13,14]. At our center, no patient has received CAR-T cell therapy after allo-HSCT due to the lack of availability in Austria. However, a total of seven patients have received bispecific antibodies with surprisingly favorable responses. Three patients with extensive extramedullary soft tissue plasmacytomas showed complete remission during therapy with teclistamab. Possibly, the allogeneic T-cell system is less exhausted, and therefore, it is possible to achieve deep and durable remissions with bispecific antibodies in such high-risk patients. 

With the approval of two BCMA-targeted CAR-T cell therapies, idecabtagene vicleucel (ide-cel) and ciltacabtagene autoleucel (cilta-cel), important alternative treatment options have recently become available, even in heavily pretreated MM patients.

Comparing the efficacy and toxicity profile of allo-HSCT and CAR-T cell therapy, CAR-T cells, and, in particular, cilta-cel, seem to provide a better outcome, especially in patients with inadequate disease control [26,27]. Regarding the toxicity profile, cilta-cel as well as allo-HSCT show a certain rate of NRM, and especially persistent neurological symptoms after CAR-T cell therapy may become a chronic burden for the patient, similarly to GVHD after allo-HSCT. Last but not least, the availability of CAR-T cell therapy is still extremely limited; thus, for well-selected, younger patients, allo-HSCT will certainly remain a therapeutic option for the treatment of RRMM in the near future.

## 5. Conclusions

Allogeneic stem cell transplantation can achieve remarkably long overall survival in heavily pretreated MM patients. To achieve optimal outcomes, strict patient selection and, especially, achieving at least VGPR before allo-HSCT are the most important factors. Harnessing the transferred, donor-derived immune-hematopoietic background after allo-HSCT, previously ineffective or intolerant therapeutics may be able to induce remissions again, and T-cell-recruiting therapies may eventually be more effective via a less-exhausted donor T-cell system. If CAR-T cell therapies become widely available, it is highly anticipated that they will be preferred over allo-HSCT, although CAR-T cell therapies also have relevant toxicities, and thoughtful patient selection will still be required.

## Figures and Tables

**Figure 1 cancers-15-05738-f001:**
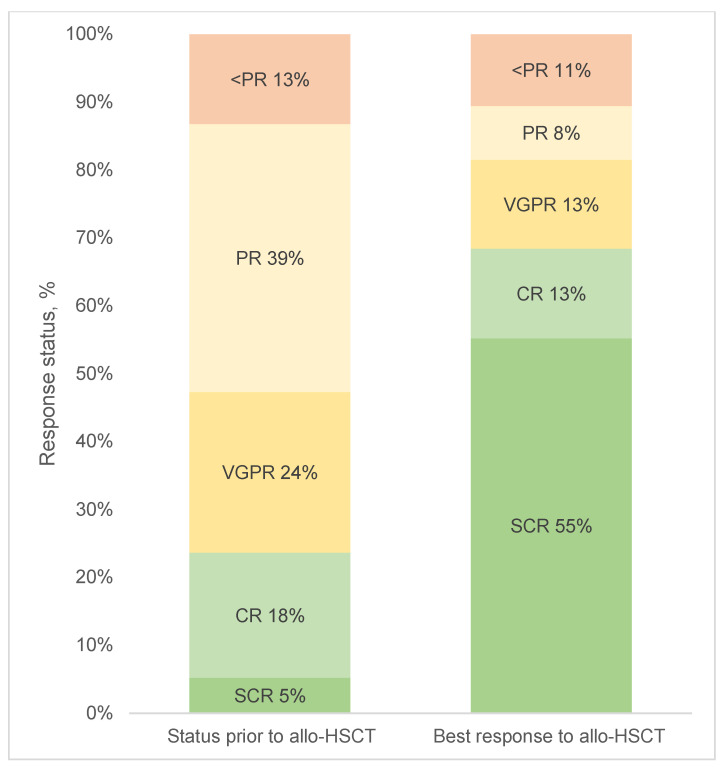
Response status of patients before and after allo-HSCT. Allo-HSCT = allogeneic stem cell transplantation, CR = complete remission, PR = partial remission, SCR = stringent complete remission, VGPR = very good partial remission.

**Figure 2 cancers-15-05738-f002:**
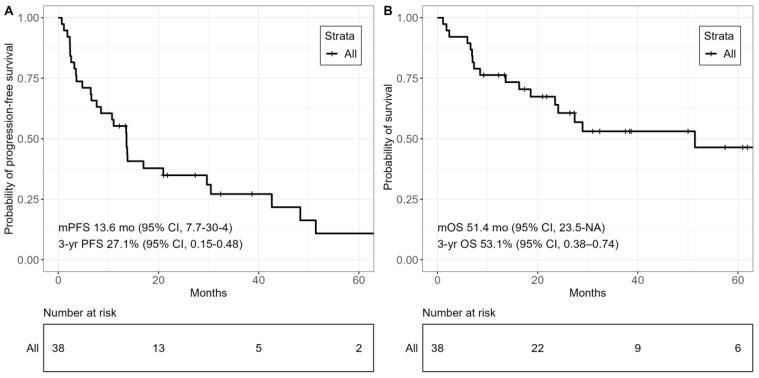
Kaplan–Meier curves of survival outcomes, overall cohort. (**A**) Progression-free survival. (**B**) Overall survival.

**Figure 3 cancers-15-05738-f003:**
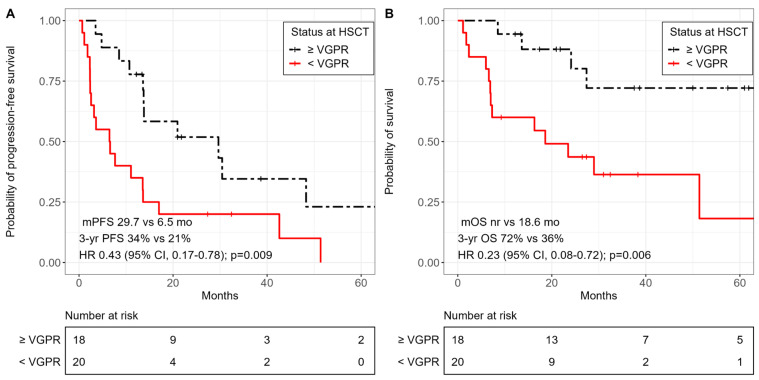
Kaplan–Meier curves of survival outcomes according to achieved response prior to allo-HSCT. (**A**) Progression-free survival. (**B**) Overall survival.

**Figure 4 cancers-15-05738-f004:**
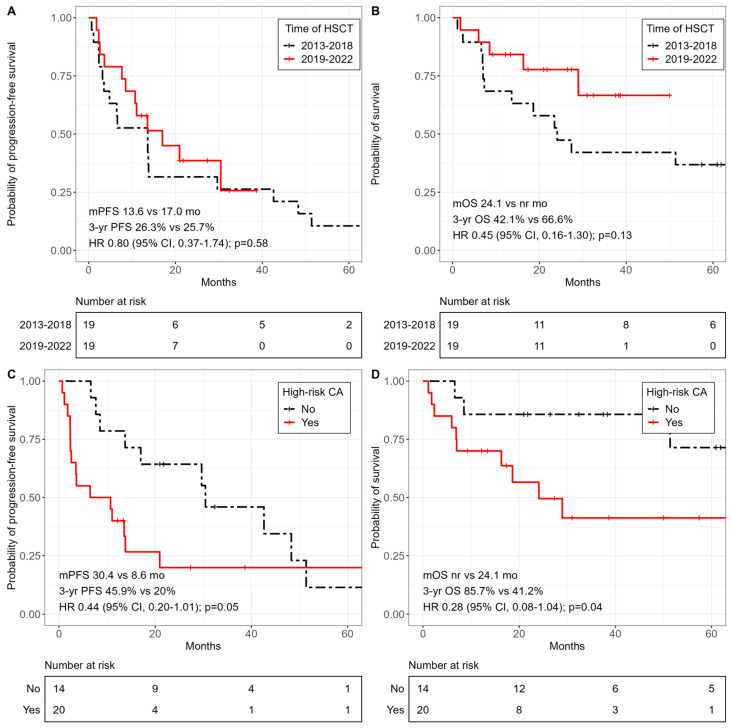
Kaplan–Meier curves of survival outcomes according to transplant date and cytogenetic risk status (**A**) Progression-free survival and (**B**) overall survival according to transplant date, 2013–2018 or 2019–2022. (**C**) Progression-free survival and (**D**) overall survival according to cytogenetic risk status, high-risk versus standard risk.

**Figure 5 cancers-15-05738-f005:**
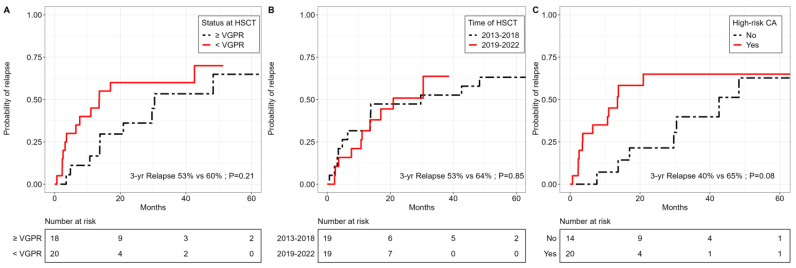
Relapse by response status prior to allo-HSCT, transplant date and cytogenetic risk status. Cumulative incidence curves for relapse according to (**A**) response of at least or less than VGPR, (**B**) transplant date between 2013 and 2018 or 2019–2022, (**C**) cytogenetic high-risk or standard-risk status.

**Figure 6 cancers-15-05738-f006:**
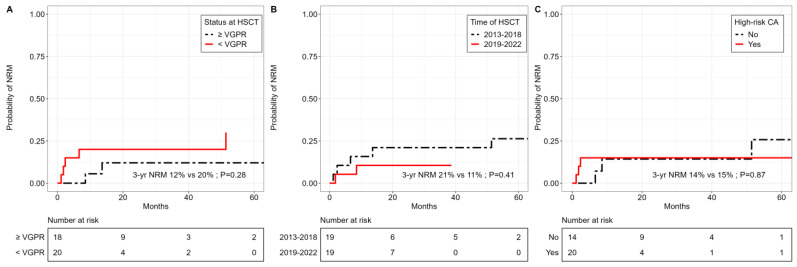
Non-relapse mortality by response status prior to allo-HSCT, transplant date and cytogenetic risk status. Cumulative incidence curves for relapse according to (**A**) response of at least or less than VGPR, (**B**) transplant date between 2013 and 2018 or 2019–2022, (**C**) cytogenetic high-risk or standard-risk status.

**Table 1 cancers-15-05738-t001:** Baseline characteristics of patients who received an allo-HSCT between 2013 and 2022.

	2013–2018 (n = 19)	2019–2022 (n = 19)	All Patients (n = 38)
Median age, years	52.8 (38.4–64.4)	58.1 (44.8–67.2)	55.2 (38.4–67.2)
Sex			
Male	12 (63%)	11 (58%)	23 (61%)
Female	7 (37%)	8 (42%)	15 (39%)
Median time from diagnosis to allo-HSCT, years	3.7 (0.6–8.2)	3.8 (0.3–13.6)	3.8 (0.3–13.6)
Extramedullary disease	2 (11%)	6 (32%)	8 (21%)
ISS stage			
I	7 (37%)	4 (21%)	11 (29%)
II	5 (26%)	5 (26%)	10 (26%)
III	3 (16%)	6 (32%)	9 (24%)
Unknown	4 (21%)	4 (21%)	8 (21%)
Cytogenetic profile			
Any high-risk CA	8 (42%)	12 (63%)	20 (53%)
High-risk IMWG	1 (5%)	5 (26%)	6 (16%)
Gain/Amp 1q	7 (37%)	10 (53%)	17 (45%)
Complex karyotype	4 (21%)	0 (0%)	4 (11%)
Median previous therapy regimens	7 (4–13)	7 (4–13)	7 (4–13)
Previous ASCT	18 (95%)	18 (95%)	36 (95%)
>1 ASCT	9 (47%)	10 (53%)	19 (50%)
Tandem ASCT	2 (11%)	3 (16%)	5 (13%)
Triple-class exposed	9 (47%)	19 (100%)	28 (74%)
Triple-class refractory	4 (21%)	5 (26%)	9 (24%)

Allo-HSCT = allogeneic stem cell transplantation, ASCT = autologous stem cell transplantation, CA = cytogenetic abnormality, IMWG = International Myeloma Working Group, ISS = International Staging System.

**Table 2 cancers-15-05738-t002:** Response at time of allo-HSCT and transplant characteristics.

	2013–2018 (n = 19)	2019–2022 (n = 19)	All Patients (n = 38)
Status at allo-HSCT			
SCR	0 (0%)	2 (11%)	2 (5%)
CR	4 (21%)	3 (16%)	7 (18%)
VGPR	5 (26%)	4 (21%)	9 (24%)
PR	8 (42%)	7 (37%)	15 (39%)
SD	0 (0%)	3 (16%)	3 (8%)
PD	2 (11%)	0 (0%)	2 (5%)
Type of transplantation			
MRD	5 (26%)	5 (26%)	10 (26%)
MUD	7 (37%)	6 (32%)	13 (34%)
Haplo	5 (26%)	7 (37%)	12 (32%)
MMUD	2 (11%)	1 (5%)	3 (8%)
Source of stem cells			
BM	6 (32%)	0 (0%)	6 (16%)
PB	13 (68%)	19 (100%)	32 (84%)
Conditioning regimen			
Myeloablative	10 (53%)	11 (58%)	21 (55%)
Reduced intensity	9 (47%)	8 (42%)	17 (45%)
Auto-allo protocol	4 (21%)	0 (0%)	4 (11%)
TBI	8 (42%)	14 (74%)	22 (58%)
GVHD prophylaxis			
ATLG	15 (79%)	10 (53%)	25 (66%)
CNI-MTX	4 (21%)	0 (0%)	4 (11%)
CNI-MMF	10 (53%)	11 (58%)	21 (55%)
PTCy-Tac-MMF	5 (26%)	8 (42%)	13 (34%)
Maintenance therapy post allo-HSCT	1 (5%)	9 (47%)	10 (26%)
DLI	8 (42%)	6 (32%)	14 (37%)

Allo-HSCT = allogeneic stem cell transplantation, ATLG = anti-T lymphocyte globulin, BM = bone marrow, CNI = calcineurin inhibitor, CR = complete remission, DLI = donor lymphocyte infusion, GVHD = graft-versus-host disease, MAC = myeloablative conditioning, MMF = mycophenolate mofetil, MMUD = mismatched unrelated donor, MRD = matched related donor, MTX = methotrexate, MUD = matched unrelated donor, NMA = non-myeloablative conditioning, PB = peripheral blood, PD = progressive disease, PR = partial remission, PTCy = post-transplant cyclophosphamide, RIC = reduced-intensity conditioning, RTC = reduced-toxicity conditioning, SCR = stringent complete remission, SD = stable disease, Tac = Tacrolimus, TBI = total body irradiation, VGPR = very good partial remission.

**Table 3 cancers-15-05738-t003:** Multivariate Cox regression analysis for overall mortality.

Variable	Hazard Ratio	*p*-Value
Less than VGPR before allo-HSCT	7.70 (1.22–48.65)	0.03
High-risk CA	5.28 (1.13–24.82)	0.04
Allo-HSCT 2013-2018	0.54 (0.12–2.38)	0.42
Interval ASCT to allo-HSCT < 2 years	2.19 (0.44–11.01)	0.34
One ASCT before allo-HSCT (versus two)	0.65 (0.15–2.86)	0.57

Allo-HSCT = allogeneic stem cell transplantation, ASCT = autologous stem cell transplantation, CA = cytogenetic abnormality, VGPR = very good partial remission.

## Data Availability

Data are available from the authors upon request.

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
