# Peer review of "Allogeneic Stem Cell Transplantation in Multiple Myeloma: Risk Factors and Outcomes in the Era of New Therapeutic Options—A Single-Center Experience"

_cancers, 2023, doi:10.3390/cancers15245738_

Round 1

Reviewer 1 Report

Comments and Suggestions for Authors

The authors present a pertinent analysis on the use of real-life allo-HSCT based on the experience of a single transplant center.

Although it is a retrospective study, the presented results are valuable regarding the role of allo-HSCT in patients with MM.

In this sense, the recommendations, based on reported results, of the authors regarding the use of allo-HSCT in a selected category of patients according to the pre-HSCT response, cytogenetic risk and the duration from ASCT to all-HSCT seem very relevant to me.

I would like to ask if there are relevant data on the evolution of transplanted patients (PFS, OS, NRM, RR) depending on the type of conditioning used and the type of transplant (MSD, MUD vs haplo).

Reviewer 2 Report

Comments and Suggestions for Authors

The authors performed unicentric retrospective analysis of myeloma patients who underwent allogeneic stem cell transplantation. The patients were 38 patients (small study), and the results were ordinary. Therefore, this manuscript has not neues. The authors described high efficacy and survival, but they have not study control. The patient selection bias affected the results of transplantation.

This manuscript has low originality, clinical significance, and scientific soundness. Therefore, overall merit for "Cancers" is low. However, the data is precious and should be accumulated. Please submit another journal.

Reviewer 3 Report

Comments and Suggestions for Authors

The authors described the outcome of 38 patients affected by multiple myeloma (MM) who underwent allogeneic SCT at their Center between 2013 and 2022. They concluded that, despite many effective new therapies avaible to MM patients, allografting still deserve a role in high risk setting.  The main limitations of this study are the retrospective nature and the number of patients, nevertheless the role of allogeneic SCT in MM remains an interesting matter of debate, and this study might add some interesting point.

I would suggest the authors some revisions:

- please add info on incidence of acute and chroinc GVHD

- I would stress that patient in less than PR at the time of allo would not gain any real advantage

- is there any differences in survival according to the number of previous treatment lines (median 7!!)

- there are too many figures and tables, I suggest to skip or put in supplementary: Figure4 EFGGH, Figure 5, Figure 6 DE, Figure 7 DE, Table 4, Table 5, Figure 8 (put in the text)

- line 290 and following please specify if you are talking about treatment at relapse post allo or as maintenance post allo

- it is interesting the info that bispecific antibodies post allo did not have additional toxicities, but the details on responses are not so informative given the small numbers.

- in the discussion line 315 and following: given the debate about the role of allo in MM, I would not talk about only treatment modality

- line 357 I would say that there are very few experiences of CAR-T in MM post allo (you can cite some abstract I believe), not surprising that you have none..

- line 363 I would not describe this single case experience in the discussion

Round 2

Reviewer 2 Report

Comments and Suggestions for Authors

The manuscript is properly revised. The data itself is precious for the future.

Reviewer 3 Report

Comments and Suggestions for Authors

The authors replied to the observations done in the first review of the manuscripta, therefore is now fine to me for publication